# Prognostic Significance of Microvessel Density and Hypoxic Markers in Canine Osteosarcoma: Insights into Angiogenesis and Tumor Aggressiveness

**DOI:** 10.3390/ani14223181

**Published:** 2024-11-06

**Authors:** Cecilia Gola, Marcella Massimini, Emanuela Morello, Lorella Maniscalco, Luiza Cesar Conti, Mariarita Romanucci, Matteo Olimpo, Leonardo Della Salda, Raffaella De Maria

**Affiliations:** 1Independent Researcher, 10100 Turin, Italy; 2Department of Veterinary Medicine, University of Teramo, 64100 Teramo, Italy; mmassimini@unite.it (M.M.); mromanucci@unite.it (M.R.); ldellasalda@unite.it (L.D.S.); 3Department of Veterinary Sciences, University of Turin, 10095 Grugliasco, Italy; emanuela.morello@unito.it (E.M.); lorella.maniscalco@unito.it (L.M.); luiza.cesarconti@unito.it (L.C.C.); matteo.olimpo@unito.it (M.O.); raffaella.demaria@unito.it (R.D.M.)

**Keywords:** canine osteosarcoma, hypoxia, angiogenesis, microvessel density, immunohistochemistry, biomarkers

## Abstract

This study investigates the role of angiogenesis, the process through which new blood vessels form, in tumor progression and clinical outcomes in canine osteosarcoma. We analyzed the relationship between microvessel density within tumors and the expression of factors associated with low oxygen levels (hypoxia) in 28 samples of appendicular canine osteosarcoma. The results revealed that specimens with a higher microvessel count were associated with a higher histological grade. Additionally, increased levels of vascular endothelial growth factor (VEGF) expression, a key regulator of blood vessel growth, were correlated with shorter disease-free interval. This suggests that VEGF may serve as a valuable marker for predicting disease progression in dogs bearing osteosarcoma. In addition, while the expression of another hypoxia-related protein, HIF-1α, showed a trend towards poorer patient survival, this was not statistically significant. These findings underscore the potential of targeting angiogenesis as part of a comprehensive treatment strategy for dogs with osteosarcoma. However, further research is necessary to gain a better understanding of the complex relationship between hypoxia, blood vessel growth, and tumor behavior, and to develop more effective therapies aimed at improving survival rates in affected dogs.

## 1. Introduction

During the progression of tumors, cancer cells require increased oxygen and nutrients, which are critical to sustain their rapid proliferation and heightened metabolic activity. As the demand for these resources grows, the expression of pro-angiogenic factors is correspondingly upregulated. However, when the rate at which tumor cells proliferate exceeds the formation of new blood vessels, hypoxic conditions arise. This hypoxic environment also promotes microenvironmental acidosis and resistance to apoptosis [1,2,3]. Additionally, tumor-associated angiogenesis often results in the development of disorganized and partially non-functional vasculature [4], further impairing the effective delivery of oxygen and nutrients to cells. In response to these unfavorable conditions, cancer cells activate signaling pathways that involve hypoxia-inducible factor (HIF-1α) as well as hypoxia-related genes, such as vascular endothelial growth factor (VEGF) [1]. This activation further stimulates angiogenesis to support tumor growth and metabolic demand [5]. This hypoxic phenotype not only fosters an aggressive and metastatic behavior, but also contributes to their resistance to therapy in several tumor types [5,6].

Angiogenesis is a key factor in cancer development, tumor growth and invasion. As such, its prognostic significance has been the focus of extensive research [7]. Numerous studies have shown a positive correlation between angiogenesis and prognosis across a variety of cancer types, including osteosarcoma (OSA) [8,9,10].

Quantifying tumor vasculature features, such as microvessel density (MVD), offers valuable insights into angiogenic activity. MVD, a quantitative measure of blood vessel formation within tumors, has emerged over the past decade as a reliable and independent predictor of overall survival (OS) and disease-free interval (DFI) in various human epithelial neoplasms, including breast, prostate, colon, and gastric cancers [11]. Higher MVD has also been associated with poorer prognosis in a limited number of canine and feline neoplasms, such as mammary tumors, squamous cell carcinoma, mast cell tumor, and soft tissue sarcomas [12,13,14]. However, in contrast to carcinomas, the relevance and predictive value of MVD in both human and canine OSA remain limited and controversial [7,15]. A recent data review and meta-analysis of MVD in human OSA patients found no significant relationship between tumoral MVD and survival parameters [16]. However, tumors that responded well to chemotherapy had higher MVD values, while those with lower MVD tumors displayed a reduced long-term DFI, suggesting that greater vascularization may support more effective adjuvant therapy. Similarly, the prognostic and clinicopathological value of VEGF remains controversial [17].

In dogs, data regarding MVD and its clinicopathological significance are currently unavailable [18]. Nevertheless, recent research has shown that hypoxia in canine OSA promotes a more aggressive tumor behavior. This is particularly evident through the association of HIF-1α expression with high-grade tumors [19]. OSA is recognized as the most common primary bone tumor in dogs [20] and it shares numerous biological and clinical similarities with its human counterpart [21], making it a significant focus of comparative oncology research. Despite advancements in treatment modalities, the prognosis for OSA remains relatively poor in both species with only limited improvement in survival rates over the past decade [22]. This underscores the crucial need for the discovery of novel biomarkers that can effectively indicate disease progression and response to treatment, which could pave the way to more effective therapies and improved outcomes.

The aim of this study is to explore the role of angiogenesis, specifically by measuring intratumoral MVD, in relation to the hypoxic marker HIF-1α and the angiogenic marker VEGF, in canine OSA. Additionally, this study will investigate the association between MVD and clinical and pathological features, as well as disease outcomes in canine OSA. By integrating these different aspects, this research aims to provide a comprehensive overview of the implications of angiogenesis in canine OSA.

## 2. Materials and Methods

### 2.1. Sample Collection and Clinical Data

A retrospective analysis was conducted on a total of 28 cases of canine appendicular OSA, which were collected at the Department of Veterinary Sciences of the University of Turin upon receiving written informed consent of the pet owner. All dogs in this study underwent limb amputation as part of their treatment protocol and subsequently received adjuvant chemotherapy, which included doxorubicin, cisplatin, and carboplatin, either as a standalone treatment or in combination. Metastatic disease in all dogs of the study cohort was excluded through thoracic radiography or computed tomography (CT). Follow-up consisted of regular clinical evaluation and thoracic radiographs, which were performed every 3 months during the first year and continued at 6-month intervals for a minimum duration of 2 years. Follow-up data were unavailable for one of the animals included in this study.

### 2.2. Histopathology

Tissues were placed in an EDTA-based decalcification solution (OsteoDec, Bio-Optica, Milan, Italy) until adequately demineralized. Formalin-fixed, paraffin-embedded (FFPE) tumor samples were routinely processed following standard laboratory protocols and stained with hematoxylin–eosin (HE) for diagnostic evaluation. The samples were then classified based on the predominant histological pattern, in accordance with the World Health Organization (WHO) guidelines [23]. Histological grading of the tumors was assessed using the Loukopoulos and Robinson grading system [24] by three independent pathologists. The authors adapted this grading system, employing the 2-tier grading approach proposed by Schott et al. [25]. This system categorizes tumors classified as grades I and II together as low-grade osteosarcomas, while tumors designated as grade III are categorized as high-grade OSAs.

### 2.3. Hypoxic Markers Immunohistochemistry

Immunohistochemistry (IHC) was conducted to assess the expression of HIF-1α (1:100 dilution; Bethyl Laboratories, Montgomery, TX, USA) and VEGF (1:25 dilution; Santa Cruz Biotechnology, Dallas, TX, USA). IHC was performed on tissue sections with a thickness of 4 μm as follows. Endogenous peroxidase activity was blocked with 0.3% H_2_O_2_ for 30 min. Then, heat-induced antigen retrieval was carried out with citrate buffer at 98 °C with a pH of 6 for 30 min. The sections were then incubated with primary antibodies overnight at 4 °C. Sections undergoing HIF-1α IHC were additionally pretreated with Triton X-100 (Sigma Aldrich, St. Louis, MO, USA) at a concentration of 0.1% for 10 min. Detection was carried out using the Vectastain Elite ABC Kit (Vector Laboratories Inc., Newark, CA, USA) with diaminobenzidine (DAB; ImmPACT DAB from Vector Laboratories Inc., Newark, CA, USA) as chromogen. Hematoxylin was used as a nuclear counterstain.

The immunolabeled slides were randomized and masked for blinded examination by independent pathologists. Established scoring systems previously described in the literature were used to evaluate the immunostaining results [19] (Table A1 and Figure 1).

### 2.4. Microvessel Density Measurement

Sections were incubated with anti-CD31 antibody (1:150 dilution; clone JC70A, Agilent Technologies, Santa Clara, CA, USA) to visualize blood endothelial cells. This step was performed following the routine immunohistochemical protocol previously outlined. The sections were then stained with periodic acid-Schiff (PAS) to highlight the basement membrane surrounding the vessels, before proceeding with hematoxylin nuclear counterstain. Briefly, the sections were incubated with 0.5% periodic acid for 5 min, and then washed with tap water and incubated with Schiff reagent Hotchkiss McManus (code 05-M20001, Bio-Optica, Milan, Italy) for 10 min. Excess dye was removed with distilled H_2_O and slides washed 3 times, each wash lasting 3 min, using a 0.5% potassium metabisulfite solution. The evaluation of MVD was performed according to the guidelines outlined in the international consensus report [26,27]. Specifically, the areas of highest neovascularization (referred to as “hotspots”) were identified at ×100 magnification. Individual microvessels were then counted on three fields at a magnification of ×200 for each sample. Any CD31-positive endothelial cell or endothelial cell cluster that was clearly separate from adjacent microvessels, tumor cells, and other connective tissue elements was considered a single, countable microvessel, in accordance with the criteria established by Spiliopoulos et al. [28]. The mean count obtained from three fields per tumor sample was expressed as MVD at magnification × 200, which was then taken into account for further analyses.

To categorize the samples based on MVD, the median MVD of all cases was used as cut-off point, thereby dividing the samples into two groups: those with low MVD (<median) and those with high MVD (≥median), in accordance with the international consensus report [27].

### 2.5. Statistical Analyses

Correlations between clinical data and the expression of hypoxic marker, assessed through IHC, as well as MVD counts were performed using the Mann–Whitney U-test and Fisher’s exact test. The Mantel–Cox log rank test and Kaplan–Meier curves were used to investigate the correlation of the clinicopathological variables with the DFI (defined as the time elapsed between surgery and the detection of metastases and/or local recurrence) and OS (defined as the time elapsed between surgery and death). Dogs that either succumbed to unrelated causes or were lost to follow-up during the study period were censored.

Data were analyzed with GraphPad Prism (version 10.2.1). A *p*-value of less than 0.05 was deemed statistically significant.

## 3. Results

### 3.1. Clinicopathological Characteristics, and Survival Outcomes in Dogs with Osteosarcoma

This study included a total of 28 dogs, with key clinicopathological features summarized in Table 1.

The age at diagnosis ranged from as young as 2 years to 13 years, with a median age of 9 years. The cohort comprised nearly equal numbers of males (*n* = 13, 46.4%) and females (*n* = 15, 53.6%). Twelve dog breeds were represented, with crossbreeds being the most common (*n* = 10, 35.71%), followed by Boxers and Rottweilers (*n* = 3 each, 10.72%), Labrador Retrievers, Czechoslovakian Wolfdogs, and German Shepherds (*n* = 2 each, 7.14%). The average weight of the dogs was approximately 34 kg (range 7.5–55 kg). The most frequently affected anatomical location was the humerus (*n* = 8, 28.57%), followed by the tibia (*n* = 6, 21.43%), femur and radius (*n* = 5 each, 17.86%).

The OSA samples were subclassified as osteoblastic (*n* = 18, 64.29%), chondroblastic (*n* = 4, 14.28%), poorly differentiated (*n* = 3, 10.72%), fibroblastic, giant-cell or telangiectatic (*n* = 1 each, 3.57%). Histological grading revealed 13 samples classified as grade III (or high-grade; 46.43%), and 12 as grade II (42.86%) and 3 as grade I (10.71%), which were collectively considered low-grade. No significant association was found among these variables.

The median DFI ranged from 29 to 1493 days with a mean of 249 days and a median of 153 days. Seventeen animals developed pulmonary metastases during the follow-up period. The median OS ranged from 36 to 1493 days with a mean of 249 days and a median of 175 days. Twenty-one animals died due to disease-related causes. Dogs that developed metastases after surgery had significantly shorter OS compared to those that did not (150 vs. 981 days; *p* = 0.01; Figure 2A). A similar statistical trend was also appreciated for DFI (115 vs. 197 days; *p* = 0.1; Figure 2B). Additionally, there was a trend toward a shorter DFI for animals affected by high-grade OSA compared to low-grade OSA (153 vs. 160 days; *p* = 0.13). No significant difference in OS was observed between tumor grades (159.5 vs. 202 days, *p* = 0.37; Figure A1).

### 3.2. Association of Microvessel Density with Tumor Grade in Canine Osteosarcoma

The median MVD for all dogs was 17 with a range of 7.67 to 75.67 (see Appendix A). For dogs with low-grade OSA, the median MVD was 15 (range of 7.67–43), whereas for those with high-grade OSA, the median MVD was 19 (range of 13.33–75.67). A significant correlation was found between hypervascularity and high-grade OSA (*p* = 0.029). When MVD was regarded as a continuous variable, a significant difference between tumor grades was observed (*p* = 0.026), with high-grade OSAs showing higher MVD than low-grade OSAs. Moreover, males showed a tendency towards higher MVD compared to females (*p* = 0.06) (Table 2). No other statistically significant associations were identified with the remaining clinicopathological parameters or hypoxic markers. Although there were minimal differences in DFI and OS between hypovascular and hypervascular tumors, these were not statistically significant (Figure A2). Interestingly, HIF-1α-positive samples showed reduced vascularity compared to negative samples; however, this difference was not statistically significant.

### 3.3. Relationship Between VEGF and HIF-1α Expression and Prognosis in Canine Osteosarcoma

High VEGF expression was significantly associated with a shorter DFI compared to low expression of this marker (106 vs. 171 days, *p* = 0.045; Figure 3B), while only a statistical trend was observed for OS (150 vs. 202 days, *p* = 0.15; Figure 3A). Nine tumor samples (32.14%) with high VEGF showed a concurrent HIF-1α positivity, while seven specimens (25%) were negative to both markers. No association was found between the two hypoxic markers (*p* = 0.43).

A total of 18 samples (64.29%) were positive for HIF-1α. Furthermore, there was a statistical trend towards indicating poorer survival times in dogs with HIF-1α-positive tumors compared to those with -negative tumors (169 vs. 727 days, *p* = 0.077). Conversely, no significant difference in DFI was observed between the two groups (153 vs. 196.5 days, *p* = 0.24; Figure A3). Additionally, a higher proportion of dogs with HIF-1α-positive tumors developed metastases over time (*n* = 13, 50%) compared to animals with HIF-1α-negative tumors (*n* = 5, 19.23%), although this difference did not reach statistical significance (*p* = 0.18). No further significant correlations were identified between the expression of hypoxic markers and other clinicopathological data.

## 4. Discussion

Cancer progression is intrinsically linked to the ability of malignant cells to secure a sufficient oxygen supply and essential nutrients for their growth. This adaptive response to high metabolic demands of rapidly dividing tumor cells is reflected in the increased angiogenesis observed in tumors [2,29]. Tumor hypoxia, which is driven by the rapid proliferation of cells, serves as a main driver of blood vessel formation. As a result, angiogenesis and its regulatory mechanisms have become a central focus of oncologic research, as they play a crucial role not only in sustaining tumor growth but also in promoting aggressive and metastatic tumor behavior [5]. In this context, the present study aims to explore the relationship between MVD, a well-established indicator of angiogenic activity, and the expression of tumor hypoxic markers in canine OSA, along with their prognostic significance.

The clinicopathological features of the population in this study largely align with the established characteristics of canine OSA. Variables such as age, breed distribution, tumor location, histopathological subtypes, and metastatic behavior observed here reflect a broader understanding of this tumor [20]. Moreover, the distribution of tumor grades, with nearly half of the cases classified as high-grade (grade III) and only a smaller proportion categorized as grade I, highlights the aggressive nature of OSA in dogs.

The observed trend for shorter DFI and OS observed in dogs diagnosed with high-grade OSA is consistent with previous reports that associated these factors with a poor prognosis and an increased metastatic rate [19]. However, it is important to note that the prognostic value of both the three-tier and the two-tier grading systems remains a subject of debate [25,30], suggesting caution in their application and interpretation. To draw firmer conclusions regarding their prognostic significance, further studies involving a larger population of canine patients, along with standardized grading criteria, are essential.

Over the last decade, MVD has been associated with poor prognosis and increased metastatic potential in several canine tumors, such as mammary gland tumors, soft tissue sarcomas and mast cell tumors [12,13,14]. These typically exhibit a strong angiogenic response, reflected in elevated MVD values, which support rapid tumor growth and dissemination. While OSA is generally considered a highly vascular tumor [18,31], this study observed comparatively lower MVD values, despite the expression of hypoxic and angiogenic factors such as VEGF. This may suggest a greater reliance on functionally mature vessels rather than on newly formed ones [31,32]. Compared to human OSA, canine OSA exhibits a relatively lower MVD [15,16,33]. The lack of correlation between MVD and VEGF expression suggests that VEGF-independent pro-angiogenic pathways may be involved [5]. Moreover, VEGF expression may influence tumor phenotype through mechanisms other than the promotion of angiogenesis, such as increasing vascular permeability [34].

This study demonstrates a significant correlation between high MVD and high-grade OSA, suggesting that increased vascularity may be a hallmark of more aggressive disease in dogs. This finding aligns with previous research in human cancers, where elevated MVD is associated with aggressive tumor behavior and increased metastatic potential [11]. However, the relationship between MVD and clinical outcomes in this study was not definitive. While tumors with higher MVD showed a tendency for shorter DFI and OS, these differences were not statistically significant. A recent review and meta-analysis of MVD in human OSA also reported a lack of correlation between MVD and survival times [16]. Speculatively, this suggests that angiogenesis may not be the sole driver of OSA development, progression and metastasis. Additionally, the bone microenvironment, rich in extracellular matrix components and featuring a unique vascular architecture, may influence metastatic pathways differently from those in soft tissue tumors. This highlights the relevance of other factors, such as bone-derived growth factors, chemokines or interaction with the bone matrix, in OSA metastasis beyond MVD alone [35,36,37].

Hypoxia represents a key feature of the tumor microenvironment of solid tumors, including canine OSA [2,19,38]. Factors such as HIF-1α and VEGF drive angiogenesis under hypoxic conditions, often contributing to aggressive tumor behavior and poor prognosis [2]. In this study, high VEGF expression was significantly associated with shorter DFI, and a similar trend was observed for OS, indicating its potential role as a prognostic marker in canine OSA. This correlation between VEGF and poor clinical outcomes mirrors findings in human OSA and other cancer types [39]. However, conflicting results in the literature [31,33] highlight the complex signaling networks regulating tumor angiogenesis. The selection of specimens based on MVD evaluation requirements may have introduced an unintentional bias, potentially explaining discrepancies with a previous study conducted by the authors [19]. Other factors, such as fibroblast growth factors (FGFs), platelet-derived growth factors (PDGFs), and angiopoietins, may also contribute to this variability in outcomes [40].

Interestingly, this study did not find a correlation between HIF-1α and VEGF expression, which suggests that VEGF regulation may involve mechanisms beyond hypoxia. These factors could include genetic mutations and alternative signaling pathways, indicating the complexity of the mechanisms governing VEGF expression [5]. This may account for the inconsistencies reported across different studies and the variability in clinical outcomes associated with these markers. Similarly, no significant correlation was observed between HIF-1α and MVD, although samples positive for the hypoxic marker showed lower MVD values than negative ones. Additionally, HIF-1α-positive OSA cases did exhibit a trend towards shorter survival times and a higher metastatic rate, although these differences were not statistically significant. This finding contrasts with a recent study where HIF-1α was associated with prognosis [19]. The discrepancies in the prognostic value of this marker may reflect differences in sample size, which could introduce a potential bias. It is worth noting that metastasis and tumor progression are multifactorial processes, with HIF-1α being one of several contributing elements [41].

## 5. Conclusions

This research makes a valuable contribution to the growing understanding of the role of angiogenesis in canine OSA, shedding light on its regulatory mechanisms and its clinicopathological significance. The findings suggest that angiogenesis holds significant potential as both a prognostic marker and a therapeutic target to improve prognosis in dogs with appendicular OSA, either as a standalone treatment or in combination with existing therapies, as seen in human studies. However, this study also brings attention to the complex interplay between vascularization, hypoxia, and disease progression in this aggressive form of cancer. It underscores the challenges and limitations of these biomarkers in fully capturing the complexity of the molecular mechanisms driving tumor hypoxia and angiogenesis. Moreover, this study acknowledges certain limitations, such as the small sample size and its retrospective design, which may introduce bias and limit the statistical power. Therefore, future studies involving larger populations and harmonized histopathologic diagnostic criteria are warranted to further explore the complex nature of OSA and the relevance of angiogenesis-associated biomarkers as therapeutic targets for canine OSA.

## Figures and Tables

**Figure 1 animals-14-03181-f001:**
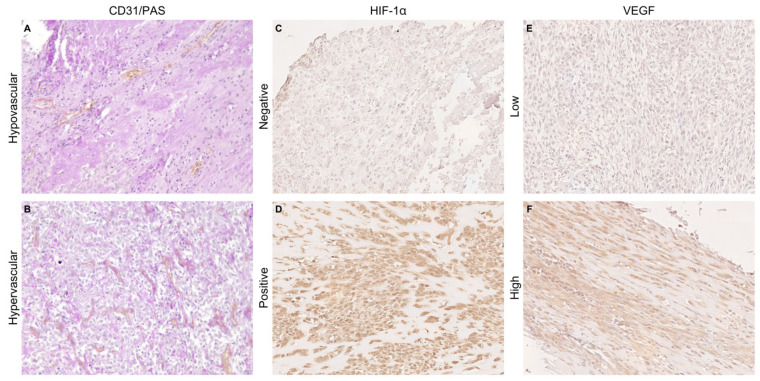
CD31/PAS double staining was used to detect tumor microvessel density of hypovascular (**A**) and hypervascular (**B**) osteosarcoma samples. Light cytoplasmic immunohistochemical staining of HIF-1α in a negative sample (**C**) compared to nuclear HIF-1α expression in a positive sample (**D**). VEGF cytoplasmic staining in samples showing low (**E**) and high expression (**F**). Immunohistochemistry (×200).

**Figure 2 animals-14-03181-f002:**
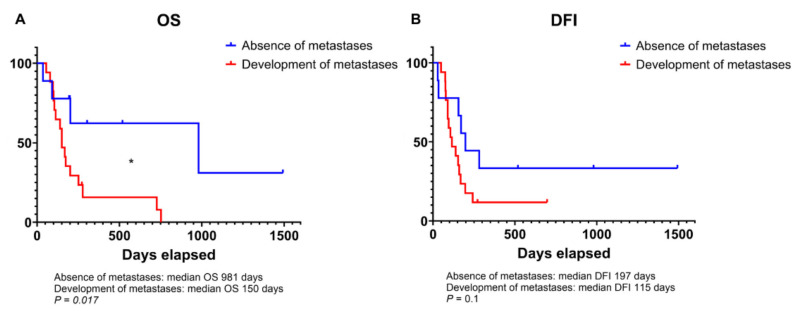
Kaplan–Meier curve illustrating the overall survival (OS; **A**) and disease-free interval (DFI; **B**) in dogs that developed metastases during follow-up compared to those that did not develop metastases. * significant *p*-value of less than 0.05.

**Figure 3 animals-14-03181-f003:**
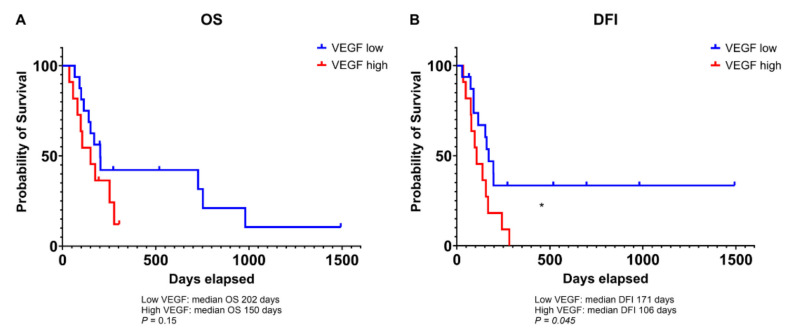
Kaplan–Meier curve illustrating the overall survival (OS; **A**) and disease-free interval (DFI; **B**) in dogs bearing osteosarcoma expressing high levels of VEGF compared to those expressing low levels. * significant *p*-value of less than 0.05.

**Table 1 animals-14-03181-t001:** Clinicopathological characteristics of the dogs included in this study.

Age (years)	Mean	8.4
	Median	9
	Range	2–13
Gender, *n* (%)	Female	13 (46.4)
	Male	15 (53.6)
Breed, *n* (%)	Crossbreed	10 (35.71)
	Boxer	3 (10.72)
	Rottweiler	3 (10.72)
	Labrador Retriever	2 (7.14)
	German Shepherd	2 (7.14)
	Czechoslovakian Wolfdog	2 (7.14)
	Other breeds	6 (21.43)
Weight (kg) ^1^	Mean	34.3
	Median	34
	Range	7.5–55
Localization, *n* (%) ^2^	Forelimb	16 (59.26)
	Hindimb	11 (40.74)
Development of metastases, *n* (%) ^1^	No	9 (34.62)
	Yes	17 (65.38)
Follow-up (days) ^3^	Mean	249
DFI	Median	153
	Range	29–1493
	Mean	293
OS	Median	175
	Range	36–1493
Histological type, *n* (%)	Osteoblastic OSA	18 (64.29)
	Chondroblastic OSA	4 (14.28)
	Poorly differentiated OSA	3 (10.72)
	Giant-cell OSA	1 (3.57)
	Fibroblastic OSA	1 (3.57)
	Telangiectatic OSA	1 (3.57)
Grading, *n* (%)	Low-grade	15 (53.57)
	High-grade	13 (46.43)

^1^ Data from two animals are missing; ^2^ data from one animal are missing; ^3^ follow-up from one animal is missing.

**Table 2 animals-14-03181-t002:** Correlation of clinicopathological features and hypoxic markers’ expression between hypovascular and hypervascular tumors.

Variable	Hypovascular MVD ≤ 16 *n* (%)	Hypervascular MVD > 16 *n* (%)	Fischer’s Exact Test *p*-Value	MVD Median (*n*)	Mann–Whitney Test *p*-Value
Sex			0.15		0.06
Male	4 (14.29)	9 (32.14)		23.33 (13)	
Female	9 (32.14)	6 (21.43)		16 (15)	
Tumor grade			*0.029*		*0.026*
Low	10 (35.71)	5 (17.86)		15 (15)	
High	3 (10.72)	10 (35.71)		19 (13)	
Development of metastases ^1^			1		0.48
No	5 (19.23)	4 (15.38)		16 (9)	
Yes	8 (30.77)	9 (34.62)		17 (17)	
HIF-1α			0.25		0.66
Negative	3 (10.72)	7 (25)		18 (10)	
Positive	10 (35.71)	8 (28.57)		16.17 (18)	
VEGF expression			1		0.81
Low	7 (25)	9 (32.14)		17 (16)	
High	6 (21.43)	6 (21.43)		15.67 (12)	

^1^ Data from two animals are missing.

## Data Availability

The original contributions presented in the study are included in this article; further inquiries can be directed to the corresponding author.

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
