# Peer review of "Prognostic Significance of Microvessel Density and Hypoxic Markers in Canine Osteosarcoma: Insights into Angiogenesis and Tumor Aggressiveness"

_animals, 2024, doi:10.3390/ani14223181_

Round 1
Reviewer 1 Report
Comments and Suggestions for Authors
Dear Authors,
I found your proposal interesting even if not particularly innovative, for which I would have some questions to ask you to try to improve your paper.
Starting from the summary, I note that you spoke at line 19 of aggressive behavior. Is this data a clinical-biological, anatomo-histopathological, instrumental data? Not to be picky, but with bone tumors these data, single or considered together, can have, and do have, a fundamental diagnostic and prognostic value.
In the abstract the main characteristics of the study are well described, but the results should be correlated, not only with the angiogenetic factors of the tumor, but also with the histotype of osteosarcoma, with the size of the tumor and of the samples examined histologically, etc. I would suggest improving these sentences with more details also in the abstract, so that whoever reads the work will have a more complete and clearer picture of the work done.
Line 107: Metastatic disease in the study cohort was excluded ... please add: IN ALL DOGS
Line 113/115: Was no decalcification performed on the investigated samples? If so, with what methodology?
The epidemiological data also appear very interesting, but I have some suggestions to make for better integration. Could you insert data relating to the trend of the degree of vascularization in relation to age? Since you describe cases from 2 to 13 years of age, could you indicate whether in younger subjects the degree of neovascularization had a similar trend or have you noticed any differences? Could you describe a comparative aspect between the degree of vascularization of a normal bone segment and the degree of neoplastic vascularization of an equal bone segment? Could you indicate whether you have detected differences in the trend of the degree of vascularization between tumors that then metastasize compared to non-metastasize?
Line 191: even if you used models referring to the prevalent histotype in your study, could you better describe if you found significant differences in neovascularization between the various forms? In addition, and above all, even if you used only 1 case of telangiectatic osteosarcoma, a tumor also characterized by an abnormal quantity of blood and vessels, could you make some differential considerations regarding the vascularization with this tumor and the others?
Finally, could you write some theories regarding the degree of parameters detected in this study and the phenomenon of metastasis. Did the most vascularized tumors metastasize before, after, more, less, no, etc.?
I believe that the F photograph relating to the immunohistochemistry of VEGF is unclear and poorly contrasted and with an overly expressed background.
Author Response
|
Response to Reviewer 1 Comments
|
||
|
|
|
|
|
The authors appreciate the reviewer’s insightful comments. We have carefully addressed the issues raised in your review and made the appropriate revisions to the manuscript. Your valuable feedback and comments have significantly contributed to improving the clarity and overall quality of the study, and we are grateful for your input. Please find the detailed responses below and the corresponding revisions highlighted in the re-submitted files. |
||
|
|
|
|
|
Point-by-point response to Comments and Suggestions for Authors |
||
|
Comment 1: Starting from the summary, I note that you spoke at line 19 of aggressive behavior. Is this data a clinical-biological, anatomo-histopathological, instrumental data? Not to be picky, but with bone tumors these data, single or considered together, can have, and do have, a fundamental diagnostic and prognostic value Response 1: The authors agree with the reviewer on the need for clarification of this term. In this context, "aggressive behavior" specifically refers to histopathological data, namely a higher histological grade. We have revised the text for clarity as follows: L 19 “We analyzed the relationship between microvessel vessel density within tumors and the expression of factors associated with low oxygen levels (hypoxia) in 28 samples of appendicular canine osteosarcoma. The findings indicate that tumors with a higher microvessel count were significantly associated with a higher histological grade, suggesting increased vascularization is linked to more aggressive tumor characteristics.” Comment 2: In the abstract the main characteristics of the study are well described, but the results should be correlated, not only with the angiogenetic factors of the tumor, but also with the histotype of osteosarcoma, with the size of the tumor and of the samples examined histologically, etc. I would suggest improving these sentences with more details also in the abstract, so that whoever reads the work will have a more complete and clearer picture of the work done. Response 2: Thank you for this helpful suggestion. We have revised the abstract to incorporate more details as recommended. The updated text now reads: L35: “Clinicopathological data such as age, breed distribution, tumor localization, histopathological sub-types, and metastatic behavior consistent with reported epidemiologic characteristics of canine OSA, though no significant correlation was found among these variables.” Due to inconsistent availability of tumor size data in our study population, this variable could not be included in the statistical analysis. Comment 3: Line 107: Metastatic disease in the study cohort was excluded ... please add: IN ALL DOGS. Response 3: The manuscript was revised accordingly (L111) Comment 4: Line 113/115: Was no decalcification performed on the investigated samples? If so, with what methodology? Response 4: The reviewer is correct; all samples underwent decalcification with Osteodec prior to processing. This detail has now been added to the Materials and Methods section. Comment 5: The epidemiological data also appear very interesting, but I have some suggestions to make for better integration. Could you insert data relating to the trend of the degree of vascularization in relation to age? Since you describe cases from 2 to 13 years of age, could you indicate whether in younger subjects the degree of neovascularization had a similar trend or have you noticed any differences? Response 5: Thank you for this insightful suggestion. Investigating the trend of vascularization relative to age is indeed valuable, not only for canine OSA but potentially for other cancers as well. In the study by Spodnick et al. (J Am Vet Med Assoc., 1992), age was identified as a significant prognostic factor, with younger dogs (<5 years) and older dogs (>10 years) showing a poorer prognosis. Based on your suggestion, we conducted additional statistical analyses to examine any potential associations between microvessel density (MVD) and age groups. However, we did not find any significant correlation between MVD and age in our cohort. Comment 6: Could you describe a comparative aspect between the degree of vascularization of a normal bone segment and the degree of neoplastic vascularization of an equal bone segment? Response 6: The authors appreciate the reviewer’s insightful question. In our study, we used Weidner’s method for microvessel density (MVD) counting (Breast Cancer Res Treat, 1995), a widely accepted technique for assessing tumor angiogenesis. According to Weidner (1995), MVD is measured by counting any brown-staining endothelial cell or cluster of endothelial cells that is distinct from adjacent microvessels, tumor cells, or connective tissue elements (N Engl J Med, 1991). We applied this method to both neoplastic tissue and adjacent normal bone, using the latter as an internal control where available. Consistent with Weidner’s findings, we observed significantly lower MVD in normal bone tissue compared to neoplastic areas. This indicates a marked increase in vascularization within the tumor, as expected in a neoplastic environment. In summary, MVD was notably higher in tumor tissue than in adjacent normal bone, supporting the enhanced angiogenic activity associated with tumor growth. Comment 7: Could you indicate whether you have detected differences in the trend of the degree of vascularization between tumors that then metastasize compared to non-metastasize? Response 7: In our cohort, we analyzed 17 samples that had metastasized and 9 samples that had not, with median microvessel density (MVD) values of 17 and 16, respectively. However, no significant differences were observed between the two groups when analyzed as both categorical and continuous variables (see Table 2). Comment 8: - Line 191: even if you used models referring to the prevalent histotype in your study, could you better describe if you found significant differences in neovascularization between the various forms? In addition, and above all, even if you used only 1 case of telangiectatic osteosarcoma, a tumor also characterized by an abnormal quantity of blood and vessels, could you make some differential considerations regarding the vascularization with this tumor and the others? Response 8: The authors appreciate the reviewer’s seek for clarity. In our study, we found that neovascularization was not statistically correlated with the prevalent histotype, which was predominantly osteoblastic. Among the histotypes, the telangiectatic osteosarcoma exhibited the second-highest mean microvessel density (MVD) at 58.67, following poorly differentiated osteosarcoma, which had an MVD of 75.67. However, it is important to note that our cohort included only one case of telangiectatic osteosarcoma. Given the limited number of cases for this and other rare histotypes, we were unable to perform a robust statistical analysis to draw meaningful conclusions regarding differences in vascularization between the various forms of osteosarcoma. Comment 9: Finally, could you write some theories regarding the degree of parameters detected in this study and the phenomenon of metastasis. Did the most vascularized tumors metastasize before, after, more, less, no, etc.? Response 9: As shown in Table 2, we did not observe a significant correlation between microvessel density (MVD) and the presence of metastasis. Although the limited number of cases restricts our ability to evaluate statistical significance comprehensively, the metastatic process in osteosarcoma is unfortunately not solely dependent on neo-angiogenesis. The literature indicates that metastasis is a complex phenomenon involving key molecules and signaling pathways related to cell proliferation, angiogenesis, and microenvironment invasion. Additionally, factors such as osteoclast activity, metabolism, immune response, and noncoding RNA signaling play critical roles in the metastatic process (Sheng et al. Front Oncol. 2021; Wilk et al. J Mol Sci. 2021). These complexities highlight the multifactorial nature of osteosarcoma metastasis and the need for further research to fully understand the interplay between these various mechanisms. |
||
|
|
||
Comment 10: I believe that the F photograph relating to the immunohistochemistry of VEGF is unclear and poorly contrasted and with an overly expressed background.
Response 10: The image has been revised

Reviewer 2 Report
Comments and Suggestions for Authors
The manuscript entitled “Prognostic Significance of Microvessel Density and Hypoxic Markers in Canine Osteosarcoma: Insights into Angiogenesis and Tumor Aggressiveness” by Cecilia Gola and colleagues presents data on the relationship between vascularization, hypoxia and osteosarcoma aggressiveness in dogs. Although the data presented are interesting because they come from the veterinary clinic, they do not ultimately provide any new information on the complex relationship between these three elements. Indeed, no significant relationship has been established between the level of expression of hypoxia markers and the degree of vascularization (Table 2), nor between vascularization and the development of metastases (Table 2), whereas the level of VEGF expression on the one hand and the presence of metastases on the other are both linked to tumor aggressiveness (Figures 3 and 2 respectively). If we take into account that the level of vascularization is significantly related to tumor grade (Table 2) and that grade is significantly related to tumor aggressiveness in terms of OS and DFI (Figure A1), there is a dichotomy here for which the authors have no explanation and which has barely been discussed in the manuscript. In particular, the heterogeneity of tumors in terms of vascularization has not been addressed, even though it is an important element! Similarly, given that vascularization is also the route of entry for chemotherapy agents, and given that the various dogs received chemotherapy treatments, the relationship between tumor vascularization level and quality of response to chemotherapy should have been studied!
The data presented in the manuscript therefore appear insufficient for publication as they stand. The reviewer suggests that the authors add a few more animals to their cohort if possible, to see if the statistically observed trends can be turned towards the significant, and to analyze their cohort in greater depth, in particular the “vascular” heterogeneity and the quality of response of the different dogs to chemotherapy.
Author Response
|
Response to Reviewer 2 Comments
|
|
|
|
|
|
The authors appreciate the reviewer’s recognition of the strengths of our study and our efforts to investigate the significance of microvessel density in canine osteosarcoma. We have carefully addressed the issues raised in your review and made the appropriate revisions to the manuscript. Your valuable feedback and insightful comments have significantly contributed to improving the clarity and overall quality of the study, and we are grateful for your input. Please find the detailed responses below and the corresponding revisions highlighted in the re-submitted files. |
|
|
|
|
|
Point-by-point response to Comments and Suggestions for Authors |
|
|
Comment 1: The manuscript entitled “Prognostic Significance of Microvessel Density and Hypoxic Markers in Canine Osteosarcoma: Insights into Angiogenesis and Tumor Aggressiveness” by Cecilia Gola and colleagues presents data on the relationship between vascularization, hypoxia and osteosarcoma aggressiveness in dogs. Although the data presented are interesting because they come from the veterinary clinic, they do not ultimately provide any new information on the complex relationship between these three elements. Response 1: We thank the reviewer for emphasizing the critical relationship between vascularization, hypoxia, and osteosarcoma aggressiveness. These factors are intricately interconnected and play a key role in cancer biology and progression, representing fundamental hallmarks of cancer. Comment 2: Indeed, no significant relationship has been established between the level of expression of hypoxia markers and the degree of vascularization (Table 2), nor between vascularization and the development of metastases (Table 2), whereas the level of VEGF expression on the one hand and the presence of metastases on the other are both linked to tumor aggressiveness (Figures 3 and 2 respectively). If we take into account that the level of vascularization is significantly related to tumor grade (Table 2) and that grade is significantly related to tumor aggressiveness in terms of OS and DFI (Figure A1), there is a dichotomy here for which the authors have no explanation and which has barely been discussed in the manuscript. In particular, the heterogeneity of tumors in terms of vascularization has not been addressed, even though it is an important element! Response 2: We appreciate the reviewer’s insights. Indeed, no statistical relationship was observed between hypoxia markers and MVD, which we acknowledge was unexpected. This result could be attributed to multiple factors, including the limited sample size, which might reduce statistical power, as well as the complex regulatory network driving hypoxia, VEGFA expression and neovascularization. While hypoxia is an important contributor to VEGFA production, it is by no means the sole driver. VEGFA expression is influenced by several pathways, such as Hedgehog signaling, NRP, and semaphorin expression, which interact in ways not solely dependent on hypoxic conditions. These factors may explain the lack of a direct association between MVD and hypoxia markers in our study. Additionally, the metastatic process and survival outcomes are inherently multifactorial, influenced by a variety of signaling pathways linked to cell proliferation, angiogenesis, immune modulation, and metabolic activity. These considerations are addressed in the discussion, In both human and canine osteosarcoma, studies have demonstrated that metastasis is associated with molecular mechanisms beyond angiogenesis, such as osteoclast activity, immunity, and non-coding RNAs (Sheng G et al., 2021; Wilk SS et al., 2021). Recent genomic studies have also found that cell proliferation pathways, rather than angiogenesis alone, are predominant in metastasis and prognosis (Das S et al., 2021; Silver KI et al., 2023). This complexity may explain why MVD and VEGFA are associated with tumor grade and survival but not directly with metastasis. Lastly, our findings align with prior research indicating that histologic grade alone does not reliably predict survival in dogs with appendicular osteosarcoma (Schott CR et al., 2018), supporting the absence of a clear link between MVD and overall survival in our cohort.
Comment 3: Similarly, given that vascularization is also the route of entry for chemotherapy agents, and given that the various dogs received chemotherapy treatments, the relationship between tumor vascularization level and quality of response to chemotherapy should have been studied! Response 3: We appreciate the reviewer’s suggestion to examine the role of vascularization in chemotherapy response, as it is indeed a critical factor in drug delivery and tumor response. In human osteosarcoma, studies have shown that tumors with higher MVD often exhibit better responses to chemotherapy, while findings indicate no association between MVD and metastasis or survival, which supports our observations (Perivoliotis et al. Cancer Invest. 2024). In canine tumors, however, a link between MVD and chemotherapy efficacy has not been identified (Mortier et al. Vet Oncology 2024), and our study similarly observed no association between MVD levels and chemotherapy response. Furthermore, all dogs in our cohort were treated using a standardized chemotherapy protocol, allowing us to measure treatment success only in terms of metastatic rates, disease-free interval (DFI), and overall survival (OS) in relation to MVD. Comment 4: The data presented in the manuscript therefore appear insufficient for publication as they stand. The reviewer suggests that the authors add a few more animals to their cohort if possible, to see if the statistically observed trends can be turned towards the significant, and to analyze their cohort in greater depth, in particular the “vascular” heterogeneity and the quality of response of the different dogs to chemotherapy. Response 4: The authors have expanded the cohort by including an additional five samples and conducted further analyses, assessing VEGFA, MVD (microvessel density), and hypoxia markers. Statistical analysis confirmed a significant association between MVD and tumor grade, as well as a relationship between VEGFA expression and metastasis with prognosis. Updated information and analyses are now included in the manuscript in the respective sections. However, no additional correlations were identified between the clinicopathological parameters, hypoxia markers, and tumor vascularization. Regarding vascular heterogeneity and the differential response to chemotherapy among the dogs, our data indicate that MVD did not significantly impact disease-free interval (DFI) or overall survival (OS) in our cases. We observed that although MVD was notably higher in high-grade tumors, it did not correlate with metastatic status. This finding suggests that while increased MVD may enhance chemotherapy delivery to the tumor, it could simultaneously facilitate be associated with aggressive tumour behaviour, balancing its effects in a way that potentially increases metastatic potential while allowing for improved chemotherapy response. Nevertheless, these insights are speculative, and the authors lack of further elements necessary to draw conclusions on this matter. Additionally, we noted that a hypoxic microenvironment could encourage the growth of dysfunctional vasculature, which may restrict both oxygen and chemotherapeutic penetration into the tumor. |
|

Round 2
Reviewer 2 Report
Comments and Suggestions for Authors
The authors have responded to the reviewer's comments as far as possible with their clinical data and have modified the manuscript in line with their responses. The reviewer has no additional comments.